# Lipid Adaptations against Oxidative Challenge in the Healthy Adult Human Brain

**DOI:** 10.3390/antiox12010177

**Published:** 2023-01-12

**Authors:** Mariona Jové, Natàlia Mota-Martorell, Èlia Obis, Joaquim Sol, Meritxell Martín-Garí, Isidre Ferrer, Manuel Portero-Otín, Reinald Pamplona

**Affiliations:** 1Department of Experimental Medicine, Lleida Biomedical Research Institute (IRBLleida), Lleida University (UdL), E-25198 Lleida, Spain; 2Catalan Institute of Health (ICS), Research Support Unit (USR), Fundació Institut Universitari per a la Recerca en Atenció Primària de Salut Jordi Gol i Gurina (IDIAP JGol), E-25007 Lleida, Spain; 3Department of Pathology and Experimental Therapeutics, University of Barcelona (UB), E-08907 Barcelona, Spain; 4Neuropathology Group, Institute of Biomedical Research of Bellvitge (IDIBELL), E-08907 Barcelona, Spain; 5Network Research Center of Neurodegenerative Diseases (CIBERNED), Instituto Carlos III, E-08907 Barcelona, Spain

**Keywords:** antioxidants, cholesterol, docosahexaenoic acid, fatty acids, lipidomics, lipid peroxidation, oleic acid, plasmalogens

## Abstract

It is assumed that the human brain is especially susceptible to oxidative stress, based on specific traits such as a higher rate of mitochondrial free radical production, a high content in peroxidizable fatty acids, and a low antioxidant defense. However, it is also evident that human neurons, although they are post-mitotic cells, survive throughout an entire lifetime. Therefore, to reduce or avoid the impact of oxidative stress on neuron functionality and survival, they must have evolved several adaptive mechanisms to cope with the deleterious effects of oxidative stress. Several of these antioxidant features are derived from lipid adaptations. At least six lipid adaptations against oxidative challenge in the healthy human brain can be discerned. In this work, we explore the idea that neurons and, by extension, the human brain is endowed with an important arsenal of non-pro-oxidant and antioxidant measures to preserve neuronal function, refuting part of the initial premise.

## 1. Introduction

Lipids are a diverse and ubiquitous group of compounds with key roles in cell physiology. The multiplicity in lipid functions is achieved by the diversity in the structures of lipid molecules [1]. The exhibited structural diversity of lipids is determined by factors such as variable acyl chain length, number and position of double bonds, head groups, and chemical changes such as oxidations, reductions, ring-forming transformations, and substitutions, as well as modification with carbohydrate residues and other functional chemical groups. There are no reliable estimates of the number of discrete lipid species in nature, but based on alkyl/acyl chain and carbohydrate permutations for glycerolipids, glycerophospholipids, and sphingolipids, the theoretical number of lipid species can be estimated as about 180,000 [2].

Interestingly, in light of available evidence, this extraordinary diversity seems not to be expressed in brain lipid composition, where species seem to be selected during evolution for their specific properties optimizing neural cell structure and functional needs. Currently, the analytical ability of profiling large-scale changes in lipid composition and determining topographical distribution of individual lipid species in neuronal and glial cells has opened a new era for the study of the neurobiology of lipids. Thus, lipidomics has revolutionized the study of lipids in neuroscience, allowing the full characterization of lipid molecular species at all levels of the biological organization (lipidome), and in any condition [3,4].

For lipid classification, the International Lipid Classification and Nomenclature Committee, on the initiative of the LIPID MAPS Consortium, developed and established a comprehensive classification system based on well-defined chemical and biochemical principles, using a framework designed to be compatible with modern informatics technology (for more details about lipid classification and nomenclature, lipid structures, and bioinformatic tools for lipidomic analysis, see [5,6,7]). Based on this classification system, lipids are currently divided into eight categories: fatty acyls (FAs); glycerolipids (GLs); glycerophospholipids (GPs); sphingolipids (SPs); saccharolipids (SLs) and polyketides (PKs); and sterol (ST) and prenol lipids (PRs), which are further divided into classes and subclasses; a unique identification is assigned to each lipid species.

The importance of lipid species in the human brain is clear, both quantitatively and qualitatively. On the one hand, lipids composed about 12% of the fresh weight and half of the dry matter of the human brain [8]. On the other hand, the structural and functional diversity of brain lipids is astonishing. From the seminal studies of the neurochemist J.L.W. Thudichum [9,10] at the turn of the 20th century, a large body of knowledge on the biology of lipids of the brain tissue has been gathered. Thus, most lipid categories, classes, and subclasses are specifically represented in neuronal and glial cells, and the spectrum of lipid molecular species can be considered as the phenotypic expression of the diverse needs and functions ascribed to them. Figure 1 shows the structure of representative lipid species present in the adult human brain, divided into their main categories.

It is assumed that the human brain is especially susceptible to oxidative stress, based on specific traits such as a higher rate of mitochondrial free radical (e.g., reactive oxygen species, ROS) production, a high content in peroxidizable fatty acids (FAs), and a low antioxidative defense (for review, see [11,12]). However, it is also evident that human neurons, although they are post-mitotic cells, survive throughout an entire lifetime. Consequently, to attenuate or avoid the impact of oxidative stress on neuron functionality and survival, they must have evolved adaptive mechanisms to cope with the deleterious effects of oxidative stress. Several of these protective features are derived from lipid adaptations (see Section 5).

**Figure 1 antioxidants-12-00177-f001:**
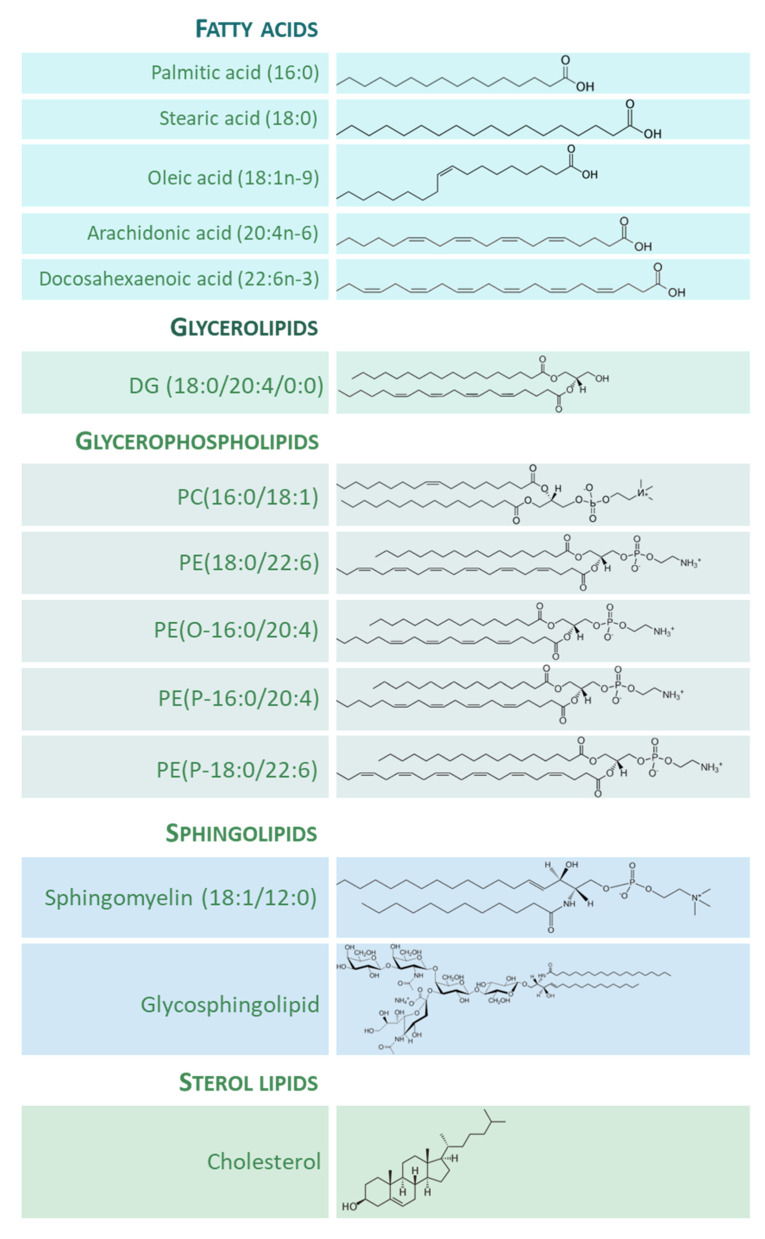
Representative lipid categories and specific molecular species present in the adult human brain. Fatty acids (FAs) shown characterize around 80–90% of total FA profile of the human brain ([13,14]). Glycerolipids (GL) in human brain comprise mono-, di-, and tri-acylglycerols. Glycerophospholipids (GP) are the major components ubiquitously found in neural cells; glycerophosphocholines, glycerophosphoethanolamines, glycerophosphoserines, and glycerophosphoinositols are the main GP present in the human brain [8]; molecular species represented are particularly abundant in human brain, highlighting the presence of ether lipids. Sphingolipids (SP) contain a common sphingoid base moiety; they are acylated to form ceramides, which are modified to generate phosphosphingolipids and glycosphingolipids. Sterol lipids comprise cholesterol and its derivatives.

This review explores the idea that neurons, glial cells (neural cells), and, by extension, the human brain, are endowed with an important arsenal of non-pro-oxidant and antioxidant measures to preserve neuronal function. We first consider a brief approach to the lipid profiles in the healthy adult human brain and how lipids affect the brain function. Then, we investigate the lipid-mediated adaptations to the physiological oxidative challenge derived from the normal brain function preserving cell integrity and survival.

## 2. Lipids in the Adult whole Human Brain

The adult whole human brain holds a large concentration of lipids with an amazing diversity of lipid classes and molecular species. The brain lipidome comprises a great diversity of GP classes and subclasses as well as a large portion of SP classes that define a specific human brain “sphingolipidome” [15]. In addition, cholesterol and its metabolites are also abundant in the human brain, containing 25% of the body’s total cholesterol [16].

GPs represent the 5% of the human whole wet brain and are the primary components found in human neural cell membranes [8]. The predominant FAs included in this lipid category are palmitic acid (16:0), palmitoleic acid (16:1n-7), stearic acid (18:0), oleic acid (18:1n-9), linoleic acid (18:2n-6), linolenic acid (18:3n-3), arachidonic acid (AA, 20:4n-6), and docosahexaenoic acid (DHA, 22:6n-3). Diacylglycerophosphates, which are precursors for GPs and neutral lipid (e.g., triradylglycerols) biosynthesis, are in low abundance in the brain (2% of total GPs). Concerning glycerophosphocholines (GPChos), the diacylglycerophosphocholines are the main form, representing 32.8% of the total content, being the primary molecular species PC (16:0-18:1) [17,18,19,20]. Among total GPChos, 2% represent ether lipid species. The glycerophosphoethanolamine (GPEtn) species account for 35.6% of total GPs [20,21]. The 1-(1Z-alkenyl),2-acylglycerophosphoethanolamines (50–60% of the GPEtn class) is the main form of this class, alkylacyl analogue content is low (5% of the GPEtn class), and diacylglycerophosphoethanolamines make up the remaining amount of GPEtns. The sn-1 glycerol position of GPEtn is mainly occupied by 16:0, 18:0, and 18:1n-9 groups; while position-2 consists of PUFA such as 20:4n-6 and 22:6n-3. The glycerophosphoserine species (GPSer) represents 16.6% of total GPs [20]. Among them, more than 90% occur as diacylglycerophosphoserines; the remaining 10% occur as 1-(1Z-alkenyl),2-acylglycerophosphoserine, containing primarily 18:0, 18:1n-9, and 22:6n-3 as FAs. Inositol phosphoglycerides account for about 2.6% of total GPs [19]. Glycerophosphoinositol (GPIn) and glycerophosphoinositol trisphosphate are additional relevant GPs with only trace amounts of glycerophosphoinositol bisphosphates. The brain contains the highest concentrations of GPIn among tissues and the main FA components are 18:0 and 20:4n-6. Finally, 0.2% of the human brain’s GPs are glycerophosphoglycerols (GPGs), and 0.1% are glycerophosphoglycerophosphoglycerols (cardiolipins) [22].

SPs are a category of complex lipids which occur in particularly large concentrations in the human brain [8,23]. This lipid category mainly consists of phosphosphingolipids (including ceramide phosphocholines (sphingomyelins), neutral glycosphingolipids (including cerebrosides of the different series: globo, ganglio, lacto, neolacto, isoglobo, mollu, and arthro), and acidic glycosphingolipids (including gangliosides and sulfoglycosphingolipids, or sulfatides), among others. Sphingomyelins account for about 14.8% of the SP content [17,20] and comprise mainly 18:0, lignoceric (24:0), and nervonic (24:1) acids. Cerebrosides amount to 15.8% of the total lipids [17]. The FA components of the cerebrosides typically contain hydroxyl FAs, and these account for more than 50% of the total FAs. Among the non-hydroxyl FAs, 24:0 and 24:1 are the main components, and cerebronic (24h:0) and hydroxynervonic (24h:1) are the most abundant hydroxyl FAs. Sulfatides are acidic glycosphingolipids and the only sulfoglycosphingolipids present in the brain, accounting for about 6.2% of the brain’s total [17]. The FA composition of sulfatides is similar to that of cerebrosides. Gangliosides, sialic acid-containing glycosphingolipids, occur ubiquitously in cell plasma membranes and are also particularly abundant in the brain [8,24]. The lipid moiety of gangliosides, the ceramide, consists of a long-chain amino alcohol linked to a FA by an amide linkage. The 18 and 20 carbon atom structures containing a trans double bond at position 4–5 are the most abundant long-chain amino alcohol components of brain gangliosides; they are generally referred to as “sphingosine”. The long-chain amino alcohols not containing a double bond at position 4–5 are generally referred to as sphinganine. Stearic acid (18:0) is the main FA of the human brain gangliosides and form over 80% of the total ganglioside FA content [24].

## 3. Functional Properties of Lipid Species in the Human Brain

Neural cell membranes are asymmetric and dynamic entities that require continuous remodeling in their lipids (chemical structure and molecular shape) in order to respond and adapt to internal and external changes. Brain lipids have different functional properties and actively contribute to guarantee the integrity of neuronal and glial cell membranes and to generate lipid messengers. Furthermore, the FAs present in these lipid species have intrinsic physicochemical properties that determine their chemical reactivity.

### 3.1. Integrity of Neural Cell Membranes

Lipids possess the inherent propensity of non-polar acyl chains to self-assemble for spontaneous membrane generation, rendering membranes virtually impermeable to polar solutes. While a single lipid species should be sufficient to generate a barrier, a cell membrane requires greater diversity and complexity in its lipid components to cover all the structural and functional needs demanded of a lipid bilayer. This diversity has implications for influencing generic chemical and physical membrane parameters such as curvature, fluidity, geometry, lipid packing density, surface charge, and thickness [25]. These properties of membrane lipids play crucial roles in the neural cell physiology, determining the regulation of receptors and ion channels, the physiology of electrical properties, synaptic transmission, etc. [25].

### 3.2. Lipid Signaling in the Human Brain

Human brain lipid species act as information-carrying molecules (for reviews, see [25,26,27,28]. In this sense, neural membrane lipids can be rapidly converted to second messengers or lipid mediators, which are lipophilic molecules that control molecular and cellular events in the brain, facilitate signal transduction processes, and regulate cell–cell communication. Lipid mediators are important endogenous regulators derived from enzymatic degradation of GP, SP, and cholesterol by phospholipases, sphingomyelinases, and cytochrome P450 hydroxylases, respectively, that can easily achieve both nuclear and organelle receptors by virtue of their chemical properties. In neural cells, lipid mediators are associated with apoptosis, differentiation, inflammation, oxidative stress, proliferation, and survival. A major group of lipid mediators, originating in the enzymatic oxidation of 20:4n-6, are referred to as eicosanoids. The corresponding lipid mediators of DHA metabolism are named docosanoids. Other GP-derived lipid mediators are diradylglycerols, diacylglycerolphosphoinositol triphosphates, platelet-activating factor, lysophosphatidic acid, and endocannabinoids. Degradation of SP also results in the generation of SP-derived lipid mediators such as ceramide, ceramide 1-phosphate, sphingosine, and sphingosine 1-phosphate. These mediators are involved in apoptosis, cell migration, differentiation, and growth. Similarly, cholesterol-derived lipid mediators, including 24- and 25-hydroxycholesterol, produce apoptosis. In general, it is interesting to highlight that the system is designed to generate redundancy, as different pathways can be used to generate the same lipid mediators. However, the full functional spectrum of lipid signaling remains to be elucidated.

### 3.3. Chemical Reactivity of the Acyl Chains in the Human Brain

Neuronal and glial cell membrane lipids are highly susceptible to oxidative damage due to the chemical reactivity of FAs [29]. The sensitivity of GP PUFA residues to oxidation (lipid peroxidation reactions) increases as the number of doble bonds per FA increases [30,31,32]. In this sense, neuronal and glial GP present a high concentration of PUFA that makes them prime targets for reaction with oxidizing agents and allows them to participate in long free radical chain reactions. Thus, lipid peroxidation generates peroxides, which can progress to the fragmentation of the FA, and produces a variety of compounds, named reactive carbonyl species (RCS), of two to nine carbons in length [33,34,35]. Subsequently, RCS can react with specific chemical functional groups in macromolecules (lipoxidation reactions) [36,37,38], resulting in the generation of a plethora of adducts and crosslinks, collectively named advanced lipoxidation end-products (ALEs) [36,39]. Using mass spectrometry and immunohistochemistry techniques, several ALEs have been detected, characterized, and located in the human brain. Table 1 offers data on ALE content in the brains of humans and different vertebrate species and shows the existence of region-specific differences in the human brain and among these vertebrate species. The consequence of ALE formation is dual. They can be detrimental to the modified cell component, as observed in most cases, but they can also act as potential signaling molecules with a neuroprotective role (see Section 5.4) [29,39].

## 4. Evolution of the Human Brain Lipid Composition

The evolution of the human brain has given rise to a structure of enormous complexity. This complexity is also reflected in the cell/tissue lipidome which, although dynamic, is strictly regulated and adapted to all biological organization levels (lipid bilayer domains, subcellular organelles, cell type, tissue, and animal species) [43,44,45,46].

Effectively, diverse studies have revealed the existence of specific traits of the human brain evolution at the lipidome level. Thus, the existence of a specific lipidome of brain tissue and a singular fingerprint of each brain region have been described [47]. This observation can be extended to other animal species (rodents and primates), in which lipidome systematically distinguish the brain from the non-neural tissues. This brain-specific lipidome [47] includes an enrichment in glycosyldiradylglycerols, GPCho, GPEtn, GPG, and neutral glycosphingolipids, and a depletion in fatty amides, triradylglycerols, and sterols. More specifically, the enriched categories are represented in specific lipid subclasses, namely ceramides, dihydroceramides, and, quite particularly, in alkenyl phosphatidylethanolamines (PE(P-) or plasmalogens) [47]. Notably, the extent of differences in the lipidome composition between the brain and non-neural tissues increases in parallel with the increase in the brain function capacity from mice to humans (Figure 2). Within the human brain, inter-regional comparative studies also demonstrated the existence of region-specific differences at the lipidome level [13,14,48]. Significantly, there is an acceleration of lipidome evolution in the neocortical regions that, in addition, specifically affects the lipid subclasses enriched in the brain. This evidence suggests that brain lipidome evolution could contribute to neocortex and brain expansion and the emergence of novel human cognitive functions [47]. Furthermore, it is postulated that lipid species played important roles during evolution that confer self-protection to the brain (see Section 5.3).

## 5. Lipid Adaptations against Oxidative Challenge in the Healthy Human Brain

It is widely accepted that the brain is highly susceptible to oxidative stress. To support this statement, different factors, such as oxygen consumption and reactive species generation, calcium, glutamate, glucose, mitochondria, generation of free radicals from an endogenous neurotransmitter metabolism, neurotransmitters’ auto-oxidation, modest endogenous antioxidant defense, microglia, redox-active transition metals, use of NOS and NOX for signaling, RNA oxidation, and unsaturated lipid enrichment, have been adduced (for review, see [11,12]). However, the observation that human neurons are functional over an entire lifetime is also irrefutable. Therefore, the existence of mechanisms that preserve function and protect against aging and age-related neurodegeneration can be inferred. In this context, it is proposed that the human brain has evolved to become resistant to stress though lipid-mediated adaptations to preserve neural cells’ integrity and cognitive function across an entire lifespan. These adaptations affect the type and content of lipids that compose neural cell membranes, and derived compounds and signaling pathways with neuroprotective properties.

### 5.1. Oleic Acid Is the Main Fatty Acid in Neural Membranes

FA are the core of most membrane lipids, such as GL, GP and SP, and play a dominant role in the physical-chemical properties of the membrane bilayer they constitute [49]. In the human brain, FAs are either saturated (SFAs), monounsaturated (MUFAs), or polyunsaturated (PUFAs) hydrocarbon chains that usually vary from 14 to 24 carbons in length, with an average chain length of 18 carbon atoms and a relative distribution, between saturated and unsaturated FAs, of around 40:60 [13,14]. Additional findings, however, demonstrate the existence of cross-regional differences in the human brain’s FA composition. Thus, there are cross-regional differences with respect to the type of unsaturated fatty acid distribution affecting, in particular, both MUFAs and PUFAs. With respect to changes in specific FA content, these findings demonstrate that the most affected FAs are, in terms of mol%, 18:1n-9, 20:1n-9, 20:4n-6, 22:4n-6, and 22:6n-3. Among these, the MUFA 18:1n-9 (oleic acid) is the most abundant FA with a range between 24 and 36% of the total FA profile (see Figure 3). These observations, verified in the gray matter from different brain regions [13,14], can also be extended to the human brain’s white matter [50].

We suggest that this oleic acid enrichment of human brain lipids expresses an evolutionary adaptation that reduces the susceptibility of cell membranes to oxidative stress while maintaining their fluidity in order to preserve the integrity and functionality of neural cells. It is known that UFA side chains (with two or more double bonds) are much more easily attacked by reactive species (e.g., free radicals) than are SFA (no double bonds) or MUFA (one double bond) side chains. In fact, the susceptibility of MUFAs, such as oleic acid, to oxidative damage is the lowest between UFAs; e.g., it is 40 times lower than a bi-unsaturated FA such as linoleic acid (18:2n-6) [29]. Therefore, oleic acid may be considered as a peroxidation-resistant FA. Consequently, the presence of a significant content of oleic acid at cell-membrane level confers to neural cells a protective role against oxidative stress present in the brain, even in physiological conditions. This evolutionary strategy allows a decrease in the sensitivity to lipid oxidation without altering membrane fluidity [51,52]. In contrast, the peroxidation index, irrespective of its position, increases with the number of double bonds [30]. Thus, the predominance of oleic acid in the FA profile provides a lesser sensitivity to lipid oxidation while maintaining fluidity, confirming that the lipid membrane is a dynamic structural adaptative system.

### 5.2. DHA-Derived Compounds Possess a Protective Role against Neural Oxidative Damage

Docosahexaenoic acid (DHA, 22:6n-3) is the most abundant PUFA in all regions of the human brain, with a mol% ranging from 2 to 13% [13,14], in line with previous observations [20,53]. This abundance in DHA content must be also interpreted in the context of the existence of inter-regional differences, and is also ascribed to different lipid molecular species, generating a region-specific lipidome [47,48].

With regards to the region-specific content in DHA, these differences are due to variances in the biosynthesis pathway which supports the inter-regional differences, and may be ascribed directly to neuronal activity, suggesting that neurons can actively maintain their own compositional profile [13]. This observation could contradict the notion that neurons do not participate actively in DHA biosynthesis and that the 22:6n-3 endowment is due to provisions proceeding from astrocytes, endothelial cells, and liver [54,55,56,57]. In this scenario, it is proposed that neurons can dynamically regulate DHA content through their ability to synthetize DHA and can actively uptake DHA from different external sources. This would be a redundant adaptive mechanism to ensure an optimal DHA pool for neuronal needs, allowing independence from potential external fluctuations, analogously to what occurs with cholesterol and plasmalogen content in the brain.

Using positron emission tomography (PET) and positron emitting tracers such as [1-11C]DHA, DHA incorporation or consumption rates in the human brain were measured as 4.6 mg/day/1500 g brain. The estimated whole brain DHA content is 5.13 g, with a half-life of 2.1 years [58]. This observation does not indicate that humans are poor DHA synthesizers, based on the very low turnover rate detected for DHA, as suggested by some authors [59,60], but, rather, highlights the importance of the preservation and extreme conservation of the DHA pool for the human brain. Indeed, the preservation of DHA in neural systems for 500–600 million years occurred despite enormous genomic changes since the beginning of animal evolution [61]. Consequently, these observations support an intimate relationship between DHA and brain function and evolution.

DHA is a key lipid with a wide spectrum of functions in neural cell biology [62,63]. DHA accomplishes essential functions ranging from structural components’ ability to quickly process events in the neural cell membrane physiology to regulation of neurogenesis, neurotransmission, and neuroprotection. The latter function is directly related to the ability of DHA to serve as precursor of a family of compounds named docosanoids. These include neuroprostanes, neurofurans, resolving Ds, protectins (neuroprotectins), and maresins. Docosanoids produce antiapoptotic, anti-inflammatory, and antioxidant effects in the brain tissue (for reviews, see [25,26,27,28]).

DHA is, however, the UFA with the highest susceptibility to lipid peroxidation (320 times higher than oleic acid). Carbonyl compounds derived from DHA oxidation, and the subsequent formation of ALEs on proteins, have been detected, identified, and quantified in different regions of the human brain, confirming DHA as a target of oxidative damage [64]. Assuming that the environment of the brain tissue is pro-oxidant, it could be suggested that the constitutive presence of DHA is a molecular suicide, and the survival of neural cells would be seriously compromised if adaptive mechanisms aimed at preventing the harmful effects derived from DHA oxidation had not been implemented during evolution. Effectively, in order to circumvent this situation, DHA has developed the ability to modulate the gene expression of the glutathione and thioredoxin antioxidant systems and related pathways in order to preserve neural cell function in the highly oxidative conditions inherent to the human brain [63].

### 5.3. Ether Lipids as Adaptive Antioxidant System in the Brain

Ether lipids constitute a GP subclass present in significant amounts in the human brain; they are the primary lipids that quantitatively and qualitatively differentiate the brain from non-neural tissues. Furthermore, brain enrichment in this lipid species has been one of the main traits of human brain evolution [8,47]. Ether lipids, in their alkyl- and alkenyl-(plasmalogen) forms, are peroxisome-derived GPs in which the acyl chain, at the sn-1 position of the glycerol backbone, is attached by an ether bond, as opposed to the ester bond in the more common diacylglycerophospholipids [65]. This seemingly simple biochemical change has profound structural and functional implications [65]. Notably, it has been suggested that plasmalogens function as endogenous antioxidants. This oxygen sensitivity of plasmalogens was described in 1972 [66,67].

The origin of aerobic life (and the subsequent generation of ROS) implied several molecular changes to guarantee a proper antioxidant defense that included, among others, the appearance and incorporation of ether lipids (plasmalogen) to eukaryotic cell membranes [68]. Interestingly, the plasmalogen biosynthesis pathway uses an oxidative mechanism that requires a source of molecular oxygen. Furthermore, the presence of a vinyl ether bond in the plasmalogen structure confers special properties to plasmalogens, including a high sensitivity to ROS, suggesting a crucial role of plasmalogens as cell free radical scavengers and constituting an important element of the endogenous antioxidant mechanism inside lipid membranes [69,70,71,72,73]. In line with this, the enrichment in plasmalogen content observed in the brain [47] could be interpreted as an adaptive response to the high oxidative conditions [11,12] while protecting unsaturated membrane lipids from oxidation by free radicals [73]. Consistent with this concept, plasmalogen-deficient cultured cells and animals are more sensitive to oxidative damage when compared to their wild-type counterparts [69,74,75,76]. In the presence of ROS, plasmalogens are quickly degraded with scission at the alkenyl ether bond [69].

### 5.4. Carbonyl Compounds Derived from Lipid Peroxidation Are Signaling Compounds which Induce Antioxidant Responses

At present, the most representative RCS are α,β-unsaturated aldehydes (acrolein and 4-hydroxy-2-nonenal (HNE)), di-aldehydes (glyoxal and malondialdehyde (MDA)), and keto-aldehydes (isoketals and 4-oxo-2-nonenal (ONE)) [35]. In addition, 2-hydroxyheptanal and 4-hydroxyhexenal are important aldehydic products of PUFA oxidation. All of them have been detected in the human brain.

Carbonyl compounds, ubiquitously generated in the human brain, have unique properties in contrast to other reactive species. For instance, compared with ROS, RCS have a much greater half-life (i.e., minutes to hours, instead of nanoseconds to microseconds for most ROS). Further, the non-charged structure of RCS allows them to migrate easily through hydrophobic membranes and hydrophilic media, thereby extending the migration distance far from the generation site and expanding the potential targets to be non-enzymatically modified [29]. In this scenario, the cytotoxicity of lipid peroxidation-derived aldehydes hinges on their abundance, half-life, reactivity, and the modified target. However, in contrast to these deleterious effects, RCS may also have regulatory function when target is a protein evolutionarily designed to sense the RCS level and to induce an adaptive neuroprotective response.

Effectively, RCS compounds have specific physiological signaling roles, inducing adaptive responses to decrease oxidative damage and enhance antioxidant defenses [29,39,77]. Two of these mechanisms involved in the prevention of oxidative damage effects in the brain are: (i) the regulation of uncoupling protein (e.g., ucp2, 4, and 5) activity resulting in a decreased mitochondrial ROS production [78,79,80,81,82] and (ii) the activation of the antioxidant response signaling (ARS) pathway mediated by the Nrf2 transcriptional factor which induces the expression of enzymes such as glutathione-S-transferase (GST), designed to detoxify carbonyl compounds [83,84,85]. As important as GST for these adaptive mechanisms is the role GPx4 (phospholipid hydroperoxide glutathione peroxidase) possesses in restoring reduced states of membrane FA from GPs to ensure lipid bilayer integrity [63,86,87,88].

### 5.5. The Non-Use of Fatty Acids by Neurons to Cover Energy Demands as Adaptive Non-Oxidant Mechanism

Neurons are especially susceptible to oxidative stress, based on specific traits such as higher mitochondrial ROS production and high UFA content. Furthermore, FAs are substrates that especially promote mitochondrial ROS generation [89,90]. These observations, taken together, suggest that the beta-oxidation pathway may be deleterious for brain mitochondria integrity [91].

Neurons are selective with respect to the substrates for energy production. Indeed, they preferentially only use glucose as fuel, while FAs are, in physiological conditions, limited. This constraint with respect to the use of glucose is not only a bioenergetics adaptation, but also a mechanism that allows neurons to switch glucose to metabolic pathways involved in the biosynthesis of antioxidants which favor neuronal survival [92]. Considering that bioenergetics performance is higher when FAs are metabolized instead of glucose, the restriction of FAs’ use by neurons appears paradoxical.

The reasons provided to explain these observations are related to the neuronal homeostasis of oxidative stress. For neurons, the use of FA as an energy substrate through mitochondrial beta-oxidation generates a series of problems [91,93]. First, ATP generation from FA demands a higher rate of oxygen consumption than glucose, thereby enhancing the risk for neurons to become hypoxic; second, the rate of ATP generation is slower than that using glucose as energy source; third, and finally, FA degradation induces a higher rate of ROS production compared to glucose, increasing oxidative stress conditions. Therefore, it can be hypothesized that the non-use of FA by neurons to cover energy demands is a non-pro-oxidant mechanism adapted during evolution to minimize the impact of oxidative stress on neuronal integrity and survival. This fact, however, does not mean that neurons abandon consuming FA if the bioenergetic conditions so require.

### 5.6. High Cholesterol Content to Regulate Oxygen Diffusion into Neural Cells

From an evolutionary point of view, the increase in atmospheric oxygen levels in an event known as the Great Oxygenation Event during the Paleoproterozoic era (1.6–2.5 billion years) imposed significant environmental pressure on primitive organisms with respect to intracellular oxygen concentration management. Lipid membranes are a natural barrier to the free transit of molecules. However, considering the size and non-polar structure of molecular oxygen, it is assumed that lipid membranes are not a restriction, or this is very reduced, to its diffusion. This non-restriction or less restricted flux of oxygen through membranes generated a biological challenge to early aerobic organisms—namely, oxidative stress. Biophysical solutions that led to highly packed lipid membranes to restrict oxygen flux arose as a possible adaptative strategy. Diverse studies have proposed that the evolutionarily selected solution was the appearance and use of sterols as a molecular adaptor for oxygen flux regulation [94,95,96]. Since oxygen availability is required for sterol synthesis, the rise in atmospheric oxygen favors the conditions for sterol generation. The result is that membrane fluidity is decreased with the incorporation of cholesterol and, consequently, oxygen diffusion. Therefore, the use of cholesterol is an evolutionary adaptation to restrict oxygen diffusion and to maintain the homeostasis of oxidative stress.

The human brain is characterized by a high demand for oxygen consumption and an extraordinary enrichment of cholesterol compared to other human tissues, suggesting that this specific cholesterol concentration is a non-random distribution. Consequently, we propose that this accumulation of cholesterol in the human brain is an adaptation to modify the lipid membrane properties of neural cells in order to restrict the intracellular diffusion of oxygen and to maintain oxidative stress to the brain within the physiological range.

## 6. Conclusions

Lipids are involved in the evolution of the human brain. The adult human brain contains a large concentration of lipids and the largest diversity of lipid classes, subclasses, and molecular species. Some of these lipids are so important for brain structure and function that neural cells have assumed the responsibility for their biosynthesis independently of the circulating content and their fluctuations. The human brain is especially susceptible to oxidative stress, based on specific traits such as a higher rate of mitochondrial ROS production, a high content in peroxidizable FA, and poor antioxidative defense. However, it is also true that human neurons, although they are post-mitotic cells, survive throughout an entire lifetime. Consequently, to reduce or avoid the challenge of oxidative stress on neuron functionality and survival, they have evolved several adaptive mechanisms to cope with the deleterious effects of oxidative stress. Several of these antioxidant features are derived from lipid adaptations (see Figure 4). In this scenario, it is proposed that neurons and, by extension, the human brain, are endowed with an arsenal of non-pro-oxidant and antioxidant lipid-derived measures to preserve neuronal integrity and function. The observations provided in this work demonstrate the importance of lipids in the evolution towards complexity and functionality of the human brain, how they become a potential source of cytotoxic compounds, and how brain tissue has developed defense mechanisms from the same lipids to protect themselves without renouncing these lipids.

## Figures and Tables

**Figure 2 antioxidants-12-00177-f002:**
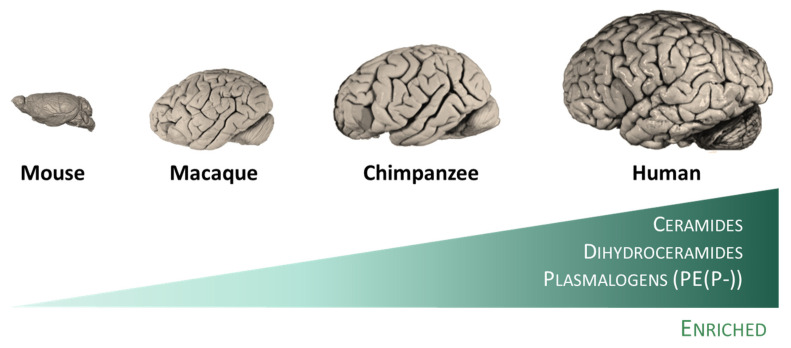
The human brain possesses specific traits at the lipidome level. Among mammalian species, humans included, lipidomes systematically distinguish the brain from the non-neural tissues [47]. This specific lipidome includes an enrichment in ceramides, dihydroceramides, and plasmalogens, and a depletion in fatty amides, triradylglycerols, and sterols. Notably, the extent of differences in the lipidome composition between the brain and non-neural tissues increases in parallel with the increase in the brain function capacity from mice to humans.

**Figure 3 antioxidants-12-00177-f003:**
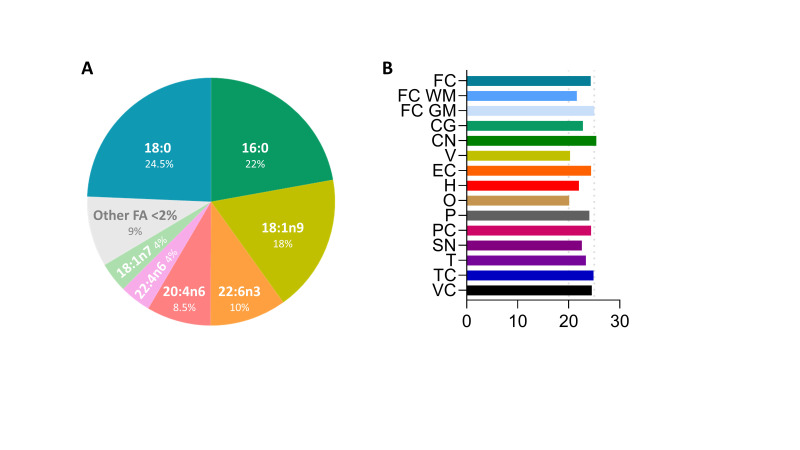
Fatty acid profile in the frontal cortex and oleic acid content (%) in different regions of the human brain. (**A**) Fatty acid distribution in human prefrontal cortex from healthy adult (middle-age) individual. Fatty acid analysis was performed in a gas chromatography system. Data obtained from [14]. (**B**) Oleic acid content (%) in different regions of the adult human brain. Data obtained from [14]. Brain regions: FC, frontal cortex; FC WM, frontal cortex white matter; FC GM, frontal cortex gray matter; CG, cingulate gyrus; CN, caudate nucleus; V, vermis; EC, entorhinal cortex; H, hippocampus; O, olive; P, putamen; PC, parietal cortex; SN, substantia nigra; T, thalamus; TC, temporal cortex; VC, visual (occipital) cortex.

**Figure 4 antioxidants-12-00177-f004:**
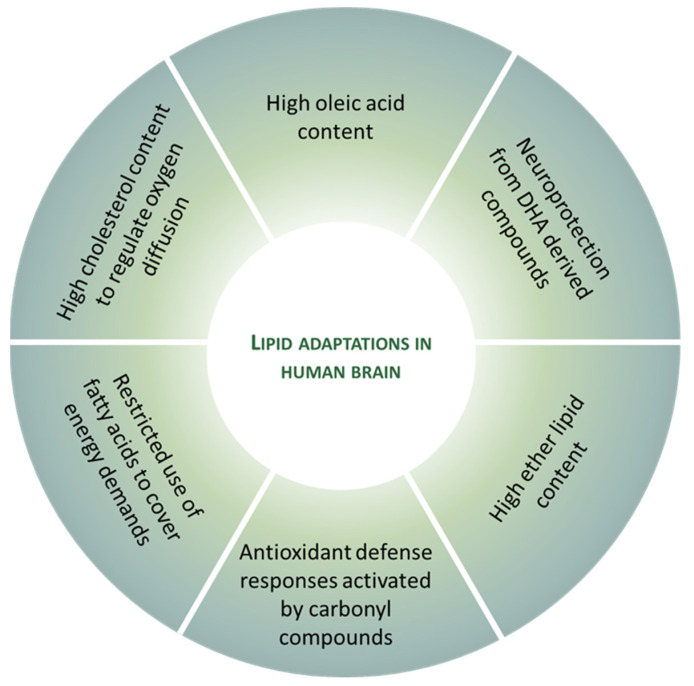
Lipid adaptations against oxidative challenge in the healthy human brain.

**Table 1 antioxidants-12-00177-t001:** Steady-state levels of the advanced lipoxidation end-product malondialdehyde-lysine (MDALys) measured by gas chromatography/mass spectrometry in distinct regions and in whole brains of different animal species, including humans.

Biological System	Animal Species	Concentration	Reference
Brain mitochondria	Rat	571 ± 30	[40]
Whole brain	Mouse	374 ± 23	[41]
Whole brain	Parakeet	305 ± 27	[41]
Whole brain	Canary	259 ± 22	[41]
Whole brain	Rat	337 ± 18	[42]
Amygdala	Human	431 ± 32	[42]
Cerebellum	Human	203 ± 20	[42]
Entorhinal cortex	Human	283 ± 28	[42]
Frontal cortex	Human	185 ± 12	[42]
Hippocampus	Human	221 ± 25	[42]
Medulla oblongata	Human	340 ± 19	[42]
Occipital cortex	Human	219 ± 16	[42]
Spinal cord	Human	352 ± 11	[42]
Striatum	Human	450 ± 52	[42]
Substantia nigra	Human	590 ± 29	[42]
Temporal cortex	Human	164 ± 9	[42]
Thalamus	Human	481 ± 42	[42]

Data are from healthy adult individuals or specimens. Values are means ± SEM. Units: µmol MDALys/mol lysine.

## Data Availability

Not applicable.

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
