# Peer review of "Lipid Adaptations against Oxidative Challenge in the Healthy Adult Human Brain"

_antioxidants, 2023, doi:10.3390/antiox12010177_

Round 1

Reviewer 1 Report

This is a carefully prepared review article.  The authors have paid attention to detail.

Author Response

R1/ This is a carefully prepared review article.  The authors have paid attention to detail.

R: We appreciate the reviewer’s comments.

Reviewer 2 Report

The information on the topic selected by the authors is abundant, but it has not yet been brought together. This manuscript is a good review on the subject matter, since it presents interesting and well-structured information. I suggest that it be published in its current format.

Author Response

R2/ The information on the topic selected by the authors is abundant, but it has not yet been brought together. This manuscript is a good review on the subject matter, since it presents interesting and well-structured information. I suggest that it be published in its current format.

R: We appreciate the reviewer’s comments.

Reviewer 3 Report

Jove and co-authors prepared a literature review and perspectives manuscript concerning the mammalian brain lipidome and protection against oxidative stress. In brief, the authors provide an overview of the various lipid species, their distribution across brain regions, and antioxidant & non-oxidant mechanisms. One compelling argument to emerge from the manuscript is that, due to its resiliency in lipid composition and antioxidant defenses, the brain parenchyma is not as vulnerable to oxidative signaling (i.e., cell death) as numerous studies throughout the literature seem to purport. Neurons in particular survive for a lifetime and there are evolutionary mechanisms that have developed to that end. See comments for clarification below.

(1)  Several "see later" indicators are in parentheses throughout the manuscript. Please specify the location of each "see later" as a defined section (e.g., see later in section 5.2).

(2) Figure 2 needs additional clarification in the legend description and primary text. Are the authors saying that fatty amides, sterols, etc. are relatively depleted in levels/role in mice versus humans? And that ceramides, etc. are significantly enriched in human brains but not at all in mice? This figure is very confusing as currently presented.

(3) The manuscript will require another thorough editing (e.g., Lines 23 & 79,..must "have" evolved...)   

Author Response

R3/ Jove and co-authors prepared a literature review and perspectives manuscript concerning the mammalian brain lipidome and protection against oxidative stress. In brief, the authors provide an overview of the various lipid species, their distribution across brain regions, and antioxidant & non-oxidant mechanisms. One compelling argument to emerge from the manuscript is that, due to its resiliency in lipid composition and antioxidant defenses, the brain parenchyma is not as vulnerable to oxidative signaling (i.e., cell death) as numerous studies throughout the literature seem to purport. Neurons in particular survive for a lifetime and there are evolutionary mechanisms that have developed to that end. See comments for clarification below.

R: We appreciate the reviewer’s comments

(1)  Several "see later" indicators are in parentheses throughout the manuscript. Please specify the location of each "see later" as a defined section (e.g., see later in section 5.2).

R: We appreciate the reviewer’s comments. In accordance with the reviewer, we have specified these indications. Please see text highlighted in yellow.

(2) Figure 2 needs additional clarification in the legend description and primary text. Are the authors saying that fatty amides, sterols, etc. are relatively depleted in levels/role in mice versus humans? And that ceramides, etc. are significantly enriched in human brains but not at all in mice? This figure is very confusing as currently presented.

R: We appreciate the reviewer’s comments. We have clarified text and legend description. Please see text highlighted in yellow (pages 6 and 7, section 4). Furthermore, Figure 2 has been modified (simplified) to avoid misinterpretation. This change makes it possible to reinforce the idea of plasmalogen enrichment at the brain level, which is relevant to the present work.

(3) The manuscript will require another thorough editing (e.g., Lines 23 & 79,..must "have" evolved...)  

R: We have corrected the sentences.